# Relationship between Perceived UX Design Attributes and Persuasive Features: A Case Study of Fitness App

**Kiemute Oyibo [1,*]** and **Julita Vassileva [2]**

1   School of Public Health Sciences, University of Waterloo, Waterloo, ON N2L 3G1, Canada
2   Department of Computer Science, University of Saskatchewan, Saskatoon, SK S7N 5C9, Canada; jiv@cs.usask.ca
*   Correspondence: kiemute.oyibo@uwaterloo.ca

**Abstract:** Research shows that a well-designed user interface is more likely to be persuasive than a poorly designed one. However, there is a limited understanding of the relationship between user-experience (UX) design attributes and users' receptiveness to the persuasive features of a persuasive technology aimed at motivating behavior change. To bridge this gap, we carried out an online case study among 228 participants from Canada and the United States to investigate the relationship between perceived UX design attributes and users' receptiveness to persuasive features. The study serves as exploratory work by focusing on a single prototype (homepage of a fitness app); four commonly researched UX design attributes (perceived aesthetics, perceived usability, perceived credibility, and perceived usefulness); and six commonly employed persuasive features (Goal-Setting/Self-Monitoring, Reward, Cooperation, Competition, Social Comparison, and Social Learning) illustrated on storyboards. The results of the Partial Least Square Path Modeling show that *perceived usefulness*, followed by *perceived aesthetics*, has the strongest relationship with users' receptiveness to the persuasive features of a fitness app. Specifically, *perceived usefulness* and *perceived aesthetics* have a significant relationship with users' receptiveness to all but two of the six persuasive features, respectively, as well as with the overall *perceived persuasiveness* of the fitness app. These findings are supported by participants' comments on the perceived UX design attributes of the fitness app and the persuasive features illustrated on the storyboards. However, *perceived usability* and *perceived credibility* have weak or non-significant relationships with users' receptiveness to the six persuasive features. The findings suggest that designers should prioritize utilitarian benefits (*perceived usefulness*) and hedonic benefits (*perceived aesthetics*) over *perceived usability* and *perceived credibility* when designing fitness apps to support behavior change.

**Keywords:** persuasive technology; persuasive feature; fitness app; goal-setting; self-monitoring; user experience; usefulness; aesthetics





## 1. Introduction

Persuasive technologies (PTs) have become pervasive in our day-to-day lives to the extent that they are becoming invisible. PTs are interactive systems intentionally designed to change attitude and behavior using the power of persuasion and social influence without deception and coercion [1]. Persuasion is the act of changing human attitudes, beliefs, and behaviors toward something through a human or an electronic means of communication. It is one of the most widely researched areas in the field of social psychology [2]. The act of persuasion is as old as humans themselves, with philosophers in ancient times such as Socrates [3] and Plato [4] using its power to persuade people to believe and accept certain ideas. In modern times, with the advancement of technology, the act of persuasion is often mediated by mass communication systems. In the traditional electronic media (such as radio and television) and, more recently, online environments (such as social media and e-commerce websites), we see the utilization of the power of PTs by

individuals and organizations in action. Moreover, in politics (through advertising) and/or business (through marketing), we see the use of persuasion and social influence in changing attitudes and eliciting certain beneficial behaviors by politicians and social influencers such as celebrities, respectively. In particular, in the physical-activity domain, due to the need to reduce the prevalence of non-communicable diseases and the incidence of aging, fitness applications have become very popular in the online marketplace [5]. However, for these fitness applications to be successful, they are equipped with persuasive features such as those proposed by Oinas-Kukkonen and Harjumaa [6] in the Persuasive System Design (PSD) model. Specifically, for these fitness applications to have the desired impact on the target audience (change negative and/or foster positive behaviors), designers equip them with personal features such as Goal-Setting, Self-Monitoring, and Reward, and social features such as Competition, Cooperation, and Social Learning [7,8]. However, in the online marketplace, including Apple Store and Google Play, potential adopters of fitness apps such as MyFitnessPal, Fitbit, and Runtastic make their decision to use them based on perceived user-experience (UX) design attributes such as *perceived aesthetics*, *perceived usability*, *perceived credibility*, and *perceived usefulness*. This makes it pertinent to understand the relationship between perceived UX design attributes in the system design domain and users' receptiveness to persuasive features in the user domain.

UX design has gained traction in human–computer interaction (HCI) research due to the growing need to not only design HCI systems that are usable but systems that are pleasing, stimulating, and enjoyable. In recent years, fitness applications have joined the plethora of persuasive applications on the market, which are intentionally designed to foster a positive user experience that can motivate users to adopt them to inform their behavior change [9–11]. On the other hand, research shows that bad user experience (e.g., poor usability) could lead to the abandonment of a mobile fitness app. For example, 25% of mobile fitness apps downloaded from the marketplace are never used again due to poor user experience [12]. Thus, in this paper, we set out to investigate the relationship between all four perceived UX design attributes of a fitness app, on one hand, and users' overall *perceived persuasiveness* and receptiveness to its six commonly employed persuasive features, on the other hand. The six persuasive features, drawn from the existing literature [13], include Goal-Setting/Self-Monitoring, Reward, Cooperation, Competition, Social Comparison, and Social Learning. Specifically, the UX design attributes are perceived by the study participants in relation to a prototyped fitness app homepage aimed at motivating behavior change. On the other hand, the six persuasive features are implemented on low-fidelity storyboards, and the study participants were asked to rate their level of receptiveness to the features and provide comments on their rating as well. We present the results of our Partial Least Square Path Modeling (PLSPM) [14], which shows the significant relationships between the perceived UX design attributes and the overall *perceived persuasiveness* of a fitness app and users' receptiveness to its persuasive features. In addition, by way of triangulation [15], we provided qualitative evidence to support the significant relationship between the perceived UX design attributes and users' receptiveness to the six persuasive features. Finally, we discussed the implications of our findings in the context of fitness app design.

The rest of the paper is organized as follows. Section 2 focuses on the background on the UX design attributes and persuasive features and related work. Section 3 describes the research method. Section 4 presents the result of our PLSPM. Section 5 focuses on the discussion of our findings. Finally, Section 6 concludes the paper.

## 2. Background and Related Work

In this section, we provide an overview on the UX design attributes and persuasive features by treating HCI systems as persuasive systems.

*2.1. UX Design Attributes*

UX is often regarded as a complex, multifaceted concept in HCI, as there is no consensus on its definition. However, there is an agreement among HCI researchers that it should not be simply equated with usability or user interface [16,17]. In particular, Følstad and Rolfsen [18] classified the UX literature into three camps relating to usability: (1) UX encompasses usability; (2) UX complements usability; and (3) UX is one of many components that constitute usability. The first camp views UX as a broad concept comprising usability among other things. For example, Petre et al. [19], in the context of e-commerce websites, viewed UX as the total customer experience which extends beyond the interaction with e-commerce products. It includes the delivery of the products, consumption of products and services, and post-sales support, all of which influence the perceptions of value and service quality and ultimately customer loyalty. The second camp views UX as an addition to the traditional notion of usability. Researchers such as Hassenzahl and Tractinsky [20] summarized UX research as a body of work that focuses on the emotional, experiential, and non-task-oriented aspects of HCI. Finally, the third camp views UX as one of the components of usability, which include effectiveness and efficiency [18].

However, owing to the advancement of HCI design beyond usable systems in recent time, with designers focusing on systems that are appealing, enjoyable, and entertaining (e.g., games), the definition of UX has been broadened to encompass the new dimension [17]. In particular, Law et al. [21] argued that the definition of UX should take a more holistic, unified approach, which encompasses the pragmatic as well as the hedonic aspects of HCI system design. The pragmatic aspects refer to the utilitarian/productive components of a HCI system design, which include usability and usefulness. On the other hand, the hedonic aspects refer to the hedonic/affective components, which include beauty and enjoyment. Hence, to cover both aspects, UX can be defined as the overall experience users derive from using or interacting with a HCI system, including how easy or pleasing it is to use the system. It is a subjective concept, making the actual experiences with HCI systems considerably different among users due to the different individual standards and from the experiences intended by the designer [22].

In the context of this paper, UX design attributes can be described as the perceived hedonic and pragmatic features of a persuasive system that help users to determine its adoption and/or use. Such attributes include *perceived aesthetics*, *perceived usability*, *perceived credibility*, and *perceived usefulness*. In the context of the Technology Acceptance Model (TAM) [10,23], these UX design attributes are conceptualized as users' cognitive beliefs about the functionality (usability and utility), beauty, and credibility of a persuasive system. Research [10,24,25] has shown that these UX design attributes (defined in Table 1) are among the key predictors of the acceptance of HCI systems. According to [26], "*UX best practices promote improving the quality of the user's interaction with and perceptions of [HCI systems and products] and any related services.*" In other words, the ultimate goal of UX is making sure that users find value in the HCI systems and products designed to meet their needs [26].

**Table 1.** UX design attributes and definition [9,25].

| Attribute | Definition |
| --- | --- |
| Perceived Aesthetics | It is the belief that a persuasive system is visually appealing and pleasing. |
| Perceived Usability | It is the belief that a persuasive system will be easy to use, understandable, and free of effort. |
| Perceived Credibility | It is the belief that a persuasive system is professionally designed and trustworthy. |
| Perceived Usefulness | It is the belief that a persuasive system possesses the required features to motivate behavior change. |

### 2.2. Persuasive Features

Persuasive features are supportive/motivational features possessed by a persuasive system, which have the capacity to motivate the behavior change of users. These persuasive features (such as Goal-Setting/Self-Monitoring, Reward, and Cooperation) and *perceived persuasiveness* are briefly defined in Table 2. The six features are drawn from the PSD model proposed by Oinas-Kukkonen and Harjumaa [13] for designing and evaluating persuasive systems. Specifically, we chose these six persuasive features because they are among those commonly studied in the literature and implemented in persuasive health applications such as fitness and healthy eating applications [7,27,28]. For example, in a systematic review of mobile applications that promoted physical activity, Matthews et al. [8] found that 70% of the applications implemented Self-Monitoring, 40% implemented Social Comparison, 25% implemented Competition, 25% implemented Social Learning, 20% implemented Reward, and 5% implemented Cooperation. Moreover, in the literature, Goal-Setting has been found to be the most vital persuasive feature for motivating individuals to pursue their health goals [26]. For example, in our prior study [29], we found that Goal-Setting/Self-Monitoring, regardless of gender or age, is the strongest predictor of users' intention to use a fitness app. We also found that Reward, Competition and Cooperation are predictors of users' intention to use a fitness app. Given that these features are commonly employed in fitness applications to motivate behavior change, it becomes important to understand how users' receptiveness to them is associated with the perceived UX design attributes of such applications.

**Table 2.** Persuasive features and definition [9,29].

| Persuasive Feature | Definition |
| --- | --- |
| Reward | Allows incentives to be awarded to users for the accomplishment of their goal. |
| Goal-Setting/ Self-Monitoring | Allows users to set goals and track their performance over time. |
| Social Learning | Allows users to observe the behaviors and achievements of other users. |
| Social Comparison | Allows users to view and compare their performance and achievements with those of others. |
| Cooperation | Allows users to work together to achieve collective goals. |
| Competition | Allows users with a common goal to compete with one another to attain it. |
| Perceived Persuasiveness | The capacity of a persuasive system to influence or motivate users to change their behavior in a positive way. |

### 2.3. Difference between UX Design Attributes and Persuasive Features

The UX design attributes of a persuasive system differ from its persuasive features. On one hand, the UX design attributes of a system refer to its perceived descriptive features (e.g., *aesthetic*, *usable*, *credible* and *useful*), which appeal to or influence users to adopt a system to accomplish a task [30]. On the other hand, persuasive features are supportive/motivational affordances (e.g., Goal-Setting, Self-Monitoring, and Cooperation), which help users to accomplish a task. Overall, persuasive features (e.g., Goal-Setting/Self-Monitoring) help users to accomplish their goals, for example, exercise regularly to become physically and mentally fit [31].

Usually, the perceived UX design attributes of a persuasive system (e.g., *perceived aesthetics*) can be evaluated quickly, for example, when users first come in contact with the homepage of a persuasive system such as a website or a fitness application. For example, Lindgaard et al. [32] found that the *perceived aesthetics* of a website is one of the key UX design attributes that grab users' attention when they first come in contact with it. According to the authors, users make the first impression about the visual appeal (*perceived aesthetics*) of a website within the first 50 milliseconds. The authors stated that the first impression about the *perceived aesthetics* of a website could influence how users judge their subsequent experience and interaction with the website. Similarly, Robins and Holmes [33]

found that the *perceived aesthetics* of a website, which has a strong influence on *perceived credibility*, is one of the key UX design attributes that may prompt a user to stay on a website or move over to another. Moreover, *perceived aesthetics* has been found to influence other perceived UX design attributes. For example, in our prior work [34] in the health domain, we found that the classical dimension of *perceived aesthetics* has a significant influence on the *perceived credibility* of a health app. Moreover, Van der Heijden [10] found that the *perceived aesthetics* of a website influences its *perceived usefulness*. On the other hand, the persuasive features of a persuasive system may require the users to carefully examine (or actually use) the system for a while to uncover its capability before deciding their intention to adopt and/or use it to motivate their behavior change.

So far, there is limited work on understanding the relationship between the perceived UX design attributes of a persuasive system and the receptiveness of its persuasive features by potential users. In our previous work on the perception of UX design attributes, we only investigated the relationship between the two main dimensions of *perceived aesthetics* and the *perceived persuasiveness* of a health app. In that study, we found that both dimensions (*classical aesthetics* and *expressive aesthetics*) have a significant influence on the *perceived persuasiveness* of a health app that featured behavior models [35]. Given the paucity of research on the relationship between the perceived UX design attributes of a fitness application and users' receptiveness to its persuasive features, we conducted a study to bridge the gap using a fitness app as a case study. The aim of the study is to investigate the significant relationship between perceived UX design attributes and users' receptiveness to the commonly employed persuasive features in PTs using a fitness app as a case study. The persuasive features, which are drawn from the PSD model, include Goal-Setting/Self-Monitoring, Reward, Cooperation, Competition, Social Comparison, and Social Learning. Overall, our work will help to uncover the perceived UX design attribute(s) that are most important in the receptiveness of the persuasive features employed in a fitness application by potential users.

## 3. Method

In this section, we present our research questions, measurement instruments, and the demographics of the study participants who took part in an online survey.

### 3.1. Research Question

In addressing the research objective, we set out to answer the following research questions:

RQ1. Is there a significant relationship between the perceived UX design attributes of a fitness application and its overall perceived persuasiveness?
RQ2. Are there significant relationships between the perceived UX design attributes of a fitness application and users' receptiveness to its persuasive features?
RQ3. Does the relationship regarding each perceived UX design attribute cut across all six persuasive features?
RQ4. How does gender moderate the relationship between perceived UX design attributes and the persuasive features?

### 3.2. Fitness App Prototype and Storyboard

To answer our research questions, we conducted a mixed-method empirical study among 228 participants using a fitness app prototype (Figure 1) and six low-fidelity storyboards (Figures A1–A6). The study is aimed to investigate the relationship between the four commonly studied UX design attributes of a fitness application, on one hand, and users' receptiveness to its overall *perceived persuasiveness* and six persuasive features commonly employed in PT design, on the other hand. The four UX design attributes of interest include *perceived aesthetics*, *perceived usability*, *perceived credibility*, and *perceived usefulness*.

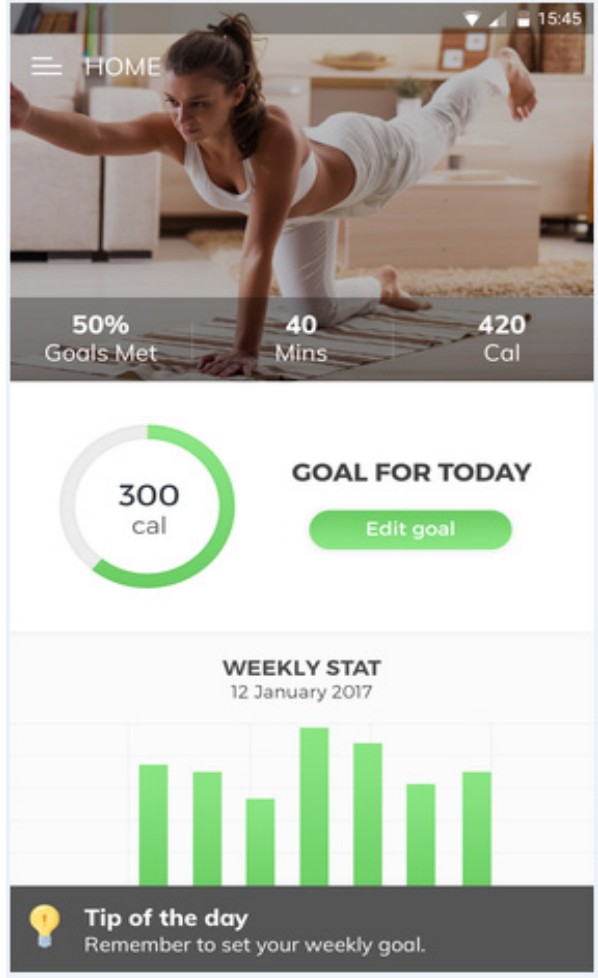

**Figure 1.** Homepage of fitness app prototype (Image of woman exercising retrieved from https://www.awaken.com/2016/09/home-yoga-practice-questions/, on 4 September 2021) [9].

The participants evaluated each of the four UX design attributes and the overall *perceived persuasiveness* of the fitness app prototype based on the homepage screen (only) shown in Figure 1. Moreover, we employed storyboards to illustrate the six persuasive features, which include Goal-Setting/Self-Monitoring, Reward, Cooperation, Competition, Social Comparison, and Social Learning. Figures A1–A6 (see the Appendix A) show screenshots of all six storyboards. The participants evaluated each of the storyboards by rating their level of receptiveness to each persuasive feature. In addition, the participants provided comments about each of the storyboards.

### 3.3. Measurement Instruments

The survey questions presented to the study participants alongside screenshots of the fitness app prototype (Figure 1) and storyboards (Figures A1–A6) illustrating the six persuasive features are shown in Table 3 Prior to presenting these questions, we provided a brief description to the participants as follows:

> *"Imagine you want to improve your personal health and fitness level. Given the challenges (e.g., time, cost, weather, etc.) associated with going to the gym regularly, the "Homex App" has been created, say by health promoters in your neighborhood, to support your physical activity."*

**Table 3.** Study's constructs and measurement items [9].

| Construct | Items in Scale |
|---|---|
| Perceived Aesthetics | 1. The app is visual.<br>2. The app is clean.<br>3. The app is pleasant.<br>4. The app is fascinating.<br>5. The app is sophisticated.<br>6. The app is creative. |
| Perceived Usability | 1. The app is easy to use.<br>2. The app is convenient to use.<br>3. The app is easy to navigate.<br>4. The app has a clear design.<br>5. The app has easy orientation. |
| Perceived Credibility | The app is credible. |
| Perceived Usefulness | 1. The app will help me improve my exercise performance.<br>2. The app will help me accomplish my exercise goals easily.<br>3. The app will be useful in my exercise.<br>4. The app will make it easier to reach my exercise goals. |
| Perceived Persuasiveness | 1. The app would influence me.<br>2. The app would be convincing.<br>3. The app would be personally relevant for me.<br>4. The app would make me reconsider my physical activity habits. |
| First Comment | Please enter here [textbox] one key feature you would expect the app to have if you were to use it. |
| Perceived Feature | Imagine that you are using the Homex App presented in the storyboard above to track your physical activity, to what extent do you agree with the following statements:<br>1. This feature of the app would influence me.<br>2. This feature of the app would be convincing.<br>3. This feature of the app would be personally relevant to me.<br>4. This feature of the app would make me reconsider my physical activity. |
| Second Comment | Provide comments about this application feature to justify your rating here [textbox]. |

Table 4 shows the scales and items in measuring the respective perceived UX design and persuasive constructs. Each item is based on a 7-point Likert scale ranging from "Strongly Disagree—1" to "Strongly Agree—7." Prior to answering the questions in Table 3, the study participants were asked in a multichoice question to study each of the storyboards and identify the correct persuasive feature illustrated from among a number of options. This question was asked to ensure that the participants studied and understood the illustrated persuasive feature to increase the reliability of their responses. Participants' responses to wrongly identified persuasive features illustrated on each of the storyboards were treated as missing data points and thus replaced by their respective average scores during the data analysis. In other words, if a respondent did not correctly identify the persuasive feature illustrated on a given storyboard, we assumed their responses to the questions regarding the storyboard are missing data points. Hence, we replace each missing data point (for each question) by the corresponding average values calculated based on the respondents who correctly identified the persuasive feature illustrated on the storyboard in question. Moreover, to measure users' receptiveness to the *perceived persuasiveness* of the fitness app and each persuasive feature, we used the adapted version of the *perceived persuasiveness* scale [36]. This scale has been validated in prior studies [37]. In addition, the participants were asked to suggest one key feature they expected the app to have if they were to use it. After the participants had finished rating the storyboards in terms of their receptiveness to the illustrated persuasive features, they were asked to provide comments to justify their

ratings. The question read, "*Provide comments about this application feature [persuasive strategy illustrated on the storyboard] to justify your rating here [textbox].*" This open question was intentionally included in the study to allow triangulation between the quantitative findings (based on PLSPM) and qualitative findings.

**Table 4.** Demographics of participants (*n* = 228).

| Variable | Subgroup | Number | Percent |
|---|---|---|---|
| Gender | Male | 132 | 57.9 |
| | Female | 95 | 41.7 |
| | Others | 1 | 0.4 |
| Age | 18–24 | 38 | 16.7 |
| | 25–34 | 122 | 53.5 |
| | 35–34 | 45 | 19.7 |
| | 45–54 | 16 | 7.0 |
| | 54+ | 7 | 3.1 |
| Education | Technical/Trade School School | 31 | 13.6 |
| | High School | 39 | 17.1 |
| | BSc | 107 | 46.9 |
| | MSc | 33 | 14.5 |
| | PhD | 6 | 2.6 |
| | Others | 2 | 0.9 |
| Country of Origin | Canada | 89 | 39.0 |
| | United States | 98 | 43.0 |
| | Others | 41 | 18.0 |
| Continent of Origin | North America | 164 | 71.9 |
| | South America | 10 | 4.4 |
| | Europe | 13 | 5.7 |
| | Africa | 11 | 4.8 |
| | Asia | 13 | 5.7 |
| | Middle East | 5 | 2.2 |
| | Others | 2 | 0.9 |

### 3.4. Participants

We submitted our study to the Research Ethics Board of our university. Upon approval, we posted it on Amazon Mechanical Turk [38] to recruit participants resident in North America (our target audience for our proposed fitness app). Each of the participants was remunerated with $1.50 in appreciation of their time. Table 4 shows the demographics of participants. Overall, 279 participants took part in the study. However, 51 participants were excluded from the data as they did not complete the survey. This resulted in 228 valid participants, comprising 132 males and 95 females.

### 3.5. Exploratory Approach

Given the limited studies in this area of research, we employed an exploratory approach to investigate the relationship between perceived UX design attributes and user' receptiveness to each of the persuasive features illustrated on the storyboards. Figure 2 shows the exploratory research model. The left-hand side comprises the perceived UX design constructs, which measure user' cognitive beliefs about the four UX design attributes, while the right-hand side comprises their belief about the overall *perceived persuasiveness*

and their receptiveness to each of the six persuasive features of a fitness app. The first research question (RQ1) is aimed at uncovering the significant and strongest relationships between the perceived UX design attributes of the fitness application (shown in Figure 1) and its overall *perceived persuasiveness*. The second research question (RQ2) is aimed at uncovering the significant relationships between the perceived UX design attributes of a fitness application and users' receptiveness to each of the persuasive features illustrated on the storyboards (Figures A1–A6). The third research question (RQ3) is aimed at uncovering which of the relationships regarding the four perceived UX design attributes cut across one or more of the six persuasive features addressed in this paper.

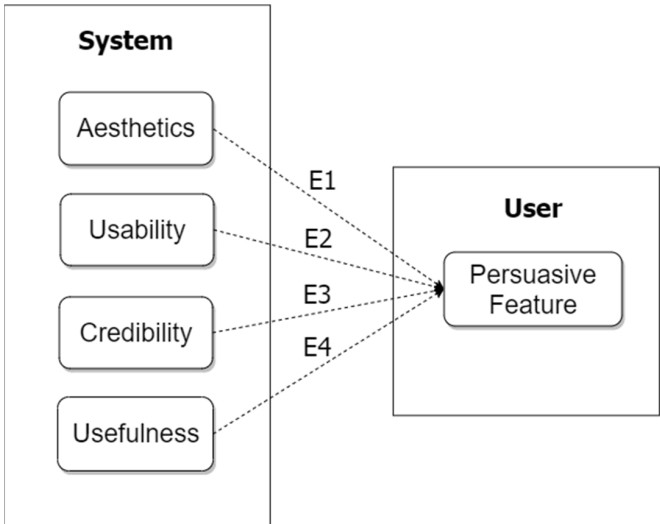

**Figure 2.** Exploratory research model.

## 4. Result

In this section, we present the results of our PLSPM, which include the evaluation of the measurement models and the analysis of the structural models. The PLSPM was carried out using the "plspm" package in R [14].

### 4.1. Measurement Models

Prior to building and analyzing a structural model, it is required that the measurement model be evaluated based on four metrics: Indicator Reliability, Internal Consistency Reliability, Convergent Validity, and Discriminant Validity. Table 5 shows the definition of each of the four preconditions and the results of the evaluation of the measurement models in the PLSPM.

**Table 5.** Evaluation of measurement models [14,39–41].

| Criterion | Definition | Evaluation Result |
|---|---|---|
| Indicator Reliability | It is the degree to which an indicator that measures a certain construct is reliable. | All of the indicators in our measurement models had an outer loading greater than 0.7. |
| Internal Consistency Reliability | It is a measure of the extent to which a set of indicators that purport to measure a certain construct produces similar scores. | This measure for each construct was evaluated using the composite reliability metric called Dillon–Goldstein (DG.rho), which was greater than 0.7. |
| Convergent Validity | It is a measure of how well the indicators that measure a certain construct are related to one another. | This criterion for each construct was evaluated using the Average Variance Extracted (AVE), which was greater than 0.5. |
| Discriminant Validity | It is a measure of the extent to which the indicators that measure a certain construct are unrelated to another construct in the measurement model. | This measure was evaluated using the crossloading metric. The results showed that no indicator loaded higher on any other construct than the one it was meant to measure. |

### 4.2. Structural Model for the Overall Perceived Persuasiveness of a Fitness App

We began our PLSPM by building the structural model of the relationship between perceived UX design attributes and the overall *perceived persuasiveness* of the fitness app, as shown in Figure 3. The path model is characterized by the goodness of fit (GOF), coefficient of determination ($R^2$), and path coefficients ($\beta$s). The GOF parameter indicates how well the model fits its data; the $R^2$ parameter captures the variance of *perceived persuasiveness*, i.e., how well the perceived UX design constructs collectively explains the variation of *perceived persuasiveness*. Finally, the $\beta$ parameter represents the strength of the relationship between each of the perceived UX design attributes and *perceived persuasiveness*. The values of GOF and $R^2$ parameters in the path model are 78% and 72%, respectively, which are considered high values in the PLSPM community [14]. Moreover, the path model shows that the strongest relationship exists between *perceived usefulness* and *perceived persuasiveness* ($\beta = 0.68$, $p < 0.001$). The second strongest relationship is that between *perceived aesthetics* and *perceived persuasiveness* ($\beta = 0.25$, $p < 0.01$). However, it turns out that *perceived credibility* has no significant relationship with *perceived persuasiveness* ($\beta = -0.09$, $p = $ n.s), while *perceived usability* has a negative relationship with *perceived persuasiveness* ($\beta = -0.13$, $p < 0.01$), which is considered weak in the PLSPM community [42].

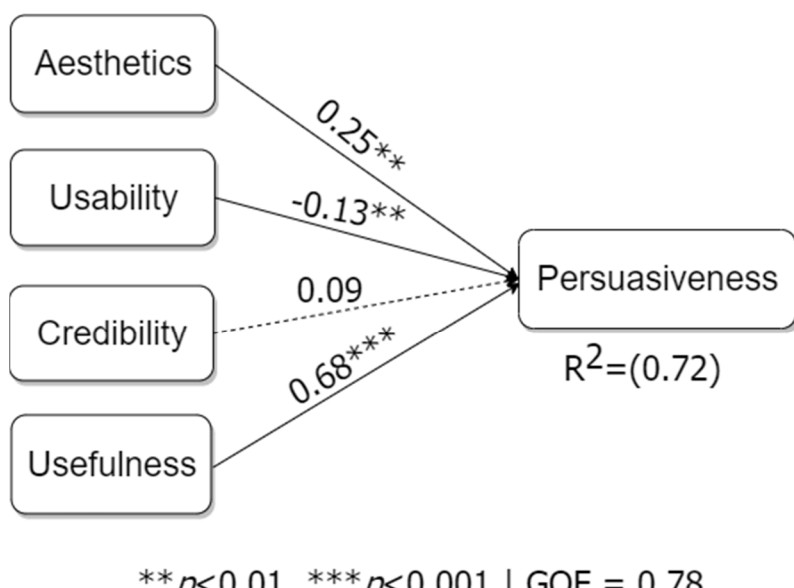

**Figure 3.** Model of the relationship between perceived UX design attributes and users' receptiveness to persuasive features of a fitness app.

### 4.3. Structural Models for the Relationship between UX Design Attributes and Persuasive Features

Figure 4 shows the path models, which depict the relationships between perceived UX design attributes and users' receptiveness to each of the six persuasive features. Regardless of the persuasive feature, the relationship between *perceived usefulness* and receptiveness to the six features is most significant, with that between *perceived usefulness* and Goal-Setting/Self-Monitoring ($\beta = 0.56$, $p < 0.001$) being the strongest, followed by that between *perceived usefulness* and Reward ($\beta = 0.53$, $p < 0.001$) and Cooperation ($\beta = 0.50$, $p < 0.001$). Moreover, the relationships between *perceived usefulness* and receptiveness to Competition ($\beta = 0.30$, $p < 0.001$) and Social Learning ($\beta = 0.22$, $p < 0.001$) are significant. Although the relationship between *perceived usability* (*perceived credibility*) and receptiveness to Cooperation (Social Learning) is significant ($\beta = -0.15$, $p < 0.05$), the relationship is negative and weak, i.e., $\beta < -0.20$, [42].

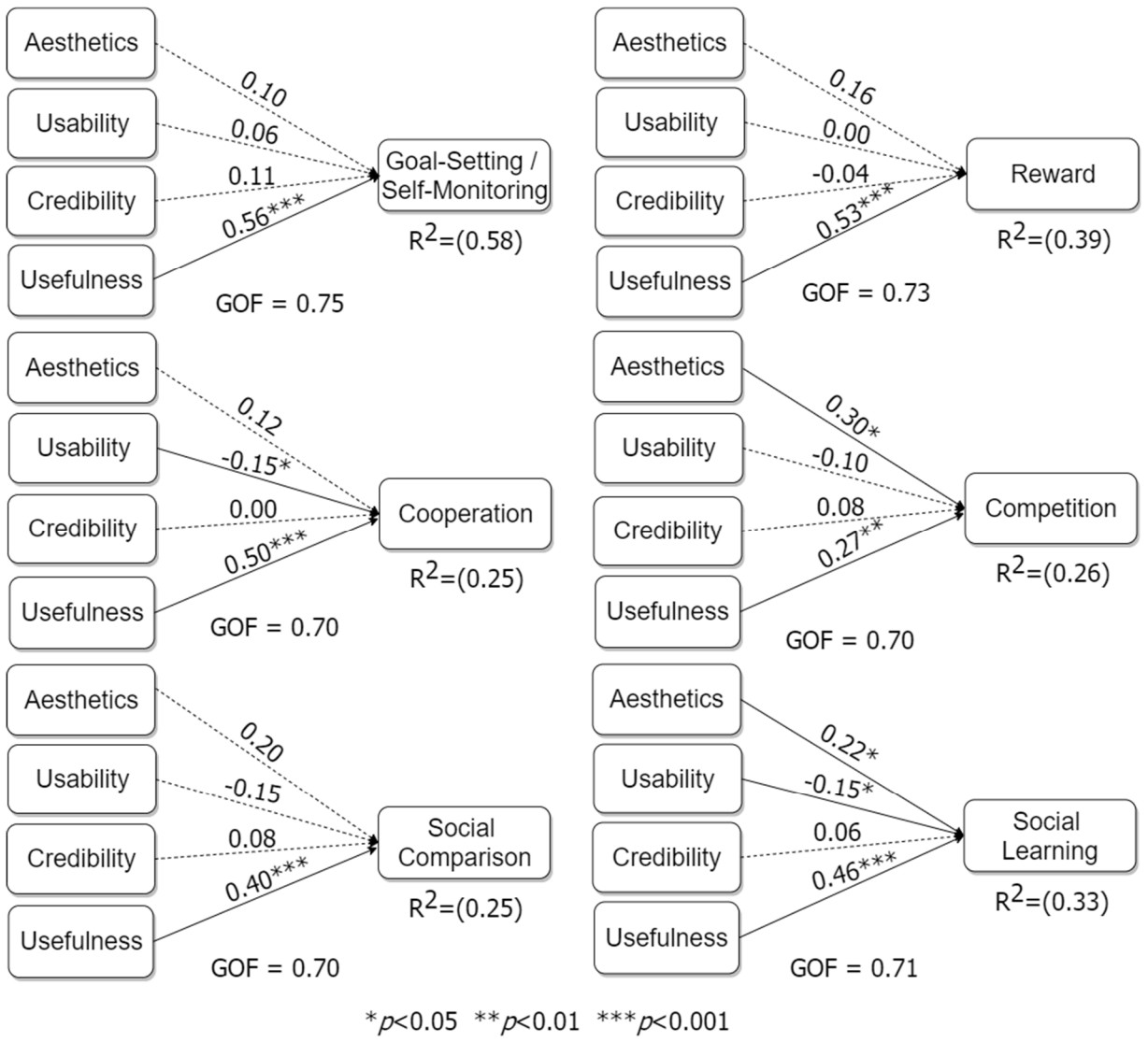

**Figure 4.** Models of the relationships between perceived UX design attributes and receptiveness to persuasive features.

### 4.4. Multigroup Analysis Based on Gender

Regarding the fourth question, Table 6 shows the results of the multigroup analyses with regard to the perceived UX design attributes, on one hand, and the overall *perceived persuasiveness* and receptiveness to the six persuasive features, on the other hand. For the gender-based models, the GOF is above 70%, and the $R^2$ values range from 26% (for male Cooperation model) to 72% (for female Perceived Persuasiveness model). The gender-based models are similar to the respective overall models (shown in Figures 3 and 4). The only significant difference between both genders is with respect to the relationship between *perceived credibility* and Cooperation ($p < 0.01$), between *perceived usefulness* and Competition ($p = 0.051$—marginal) and between *perceived credibility* and Social Learning ($p = 0.055$—marginal). We discuss the gender differences in detail in Section 5.3.

**Table 6.** Gender-based multigroup analysis. SMT = Self-Monitoring, sig = significance, n.s = non-significant, * $p < 0.05$, ** $p < 0.01$, *** $p < 0.001$.

| Relationship | Overall | Male | Female | *p*-Value | Sig |
|---|---|---|---|---|---|
| Aesthetics→Persuasiveness | 0.25 ** | 0.31 * | 0.23 * | 0.358 | n.s |
| Usability→Persuasiveness | −0.13 ** | −0.15 * | −0.13 | 0.257 | n.s |
| Credibility→Persuasiveness | 0.09 | 0.10 | 0.11 | 0.261 | n.s |
| Usefulness→Persuasiveness | 0.68 *** | 0.64 *** | 0.72 ** | 0.373 | n.s |
| R$^2$ | 0.72 | 0.74 | 0.76 | - | - |
| Aesthetics→Goal-Setting/SMT | 0.10 | 0.12 | 0.00 | 0.392 | n.s |
| Usability→Goal-Setting/SMT | 0.06 | 0.07 | 0.14 | 0.293 | n.s |
| Credibility→Goal-Setting/SMT | 0.11 | 0.18 | 0.18 | 0.183 | n.s |
| Usefulness→Goal-Setting/SMT | 0.56 *** | 0.53 *** | 0.51 *** | 0.264 | n.s |
| R$^2$ | 0.58 | 0.65 | 0.53 | - | - |
| Aesthetics→Reward | 0.16 | 0.26* | 0.06 | 0.097 | n.s |
| Usability→Reward | 0.00 | 0.04 | 0.05 | 0.254 | n.s |
| Credibility→Reward | −0.04 | −0.09 | 0.10 | 0.435 | n.s |
| Usefulness→Reward | 0.53 *** | 0.45 ** | 0.54 *** | 0.208 | n.s |
| R$^2$ | 0.39 | 0.41 | 0.46 | - | - |
| Aesthetics→Cooperation | 0.12 | 0.00 | 0.18 | 0.321 | n.s |
| Usability→Cooperation | −0.15 * | −0.07 | −0.09 | 0.195 | n.s |
| Credibility→Cooperation | 0.00 | 0.11 | −0.06 | 0.005 | yes |
| Usefulness→Cooperation | 0.50 *** | 0.48 *** | 0.53 *** | 0.271 | n.s |
| R$^2$ | 0.25 | 0.26 | 0.32 | - | - |
| Aesthetics→Competition | 0.30 * | 0.39 * | 0.15 | 0.145 | n.s |
| Usability→Competition | −0.10 | 0.01 | −0.05 | 0.224 | n.s |
| Credibility→Competition | 0.08 | 0.06 | 0.05 | 0.276 | n.s |
| Usefulness→Competition | 0.27 ** | 0.18 | 0.45 *** | 0.051 | marginal |
| R$^2$ | 0.26 | 0.34 | 0.29 | - | - |
| Aesthetics→Social Comparison | 0.20 | 0.15 | 0.25 | 0.443 | n.s |
| Usability→Comparison | −0.15 | −0.01 | −0.18 | 0.285 | n.s |
| Credibility→Comparison | 0.08 | 0.12 | 0.02 | 0.393 | n.s |
| Usefulness→Comparison | 0.40 *** | 0.38 ** | 0.45 ** | 0.498 | n.s |
| R$^2$ | 0.25 | 0.33 | 0.28 | - | - |
| Aesthetics→Social Learning | 0.22 * | 0.11 | 0.28 * | 0.12 | n.s |
| Usability→Social Learning | −0.15 * | −0.11 | −0.08 | 0.16 | n.s |
| Credibility→Social Learning | 0.06 | 0.20 * | 0.02 | 0.055 | marginal |
| Usefulness→Social Learning | 0.46 *** | 0.45 ** | 0.50 * | 0.218 | n.s |
| R$^2$ | 0.33 | 0.36 | 0.45 | - | - |

Table 7 shows a summary of the relationship between perceived UX design attributes and users' receptiveness to persuasive features for the entire dataset. The main takeaway of this summary is that the relationship between *perceived usefulness* and all of the persuasive features (including that of the overall perceived persuasiveness of the fitness app) are significant at $p < 0.05$. The second takeaway is that the relationship between *perceived*

*aesthetics*, on one hand, and Competition/Social Learning/*perceived persuasiveness*, on the other hand, is significant as well at $p < 0.05$.

**Table 7.** Summary of the relationship between perceived UX design attributes and users' receptiveness to persuasive features. ✔ = positive significant relationship at $p < 0.05$, - = negative signiificant relationship at $p < 0.05$, × = non-significant relationship.

| Relationship | Aesthetics | Usability | Credibility | Usefulness |
|---|---|---|---|---|
| Perceived Persuasiveness | ✔ | - | × | ✔ |
| Goal-Setting/Self-Monitoring | × | × | × | ✔ |
| Reward | × | × | × | ✔ |
| Cooperation | × | - | × | ✔ |
| Competition | ✔ | × | × | ✔ |
| Social Comparison | × | × | × | ✔ |
| Social Learning | ✔ | - | × | ✔ |

*4.5. Sample Comments Supporting the Significant Relationship between Perceived UX Design Attributes and Users' Receptiveness to Persuasive Features of a Fitness App*

In addition to the PLSPM, we manually went through the participants' comments to uncover qualitative evidence that supports the significant relationship between the perceived UX design attributes and users' receptiveness to the persuasive features of a fitness app. Tables 8 and 9 show a cross-section of the participants' comments supporting the relationship between *perceived usefulness* and users' receptiveness to the six persuasive features. Specifically, Table 8 focuses on the participants who rated the *perceived usefulness* of the fitness app high and their receptiveness to its six persuasive features high as well.

On the other hand, Table 9 focuses on the participants who rated the *perceived usefulness* of the fitness app low and their receptiveness to the six persuasive features low as well. For example, in Table 8, P49 perceived the fitness app as highly *useful* (M = 5.5/7) and, as a result, was highly receptive to the Cooperation feature (M = 7/7). With regard to the key feature she might want the fitness app to have, she requested an app that supports "*exercises that will reach my goal.*" Similarly, the participant commented on the Cooperation feature; thus, "*If someone is relying on me then I will do it [engage in exercise].*" Moreover, Tables 10 and 11 show a cross-section of the participants' comments supporting the relationship between *perceived aesthetics* and users' receptiveness to the six persuasive features. Table 10 is for high-raters of both *perceived aesthetics* and receptiveness to the persuasive features. On the other hand, Table 11 is for low-raters of both *perceived aesthetics* and receptiveness to the persuasive features. For example, in Table 10, P22, who perceived the fitness app as highly *aesthetic* (M = 5.12/7), was highly receptive to the Competition feature (M = 7/7). Thus, she commented that Competition "*is another good way to motivate me to use the app.*"

**Table 8.** Sample comments on the relationship between perceived usefulness and receptiveness to persuasive features. GST/SMT = Goal-Setting/Self-Monitoring, REWD = Rewards, COOP = Cooperation, CMPT = Competition, SCOMP = Social Comparison, SLEARN = Social Learning.

| No. | Usefulness | Key Feature | | Persuasive Feature |
|---|---|---|---|---|
| [P27, M] | 7 | "*Scheduled exercise routine*" | GST/SMT = 6 | "*I like how it informs me of what I need to improve in order to meet my goals.*" |
| [P14, F] | 6.75 | "*I would like the app to remind me to drink water.*" | GST/SMT = 7 | "*It would make me want to work in my goal and in the calories I should burn*" |
| [P216, F] | 6.5 | "*The ability to add friends for support.*" | REWD = 7 | " . . . .*I would want to reach the goal and earn the points.*" |

**Table 8.** *Cont.*

| No. | Usefulness | Key Feature | | Persuasive Feature |
|---|---|---|---|---|
| [P49, F] | 5.5 | *"Exercises that will help reach my goal."* | REWD = 6 | *"In a way this is competitive because I am competing to win something—which works well for me."* |
| [P49, F] | 5.5 | *"Exercises that will help reach my goal."* | COOP = 7 | *"If someone is relying on me then I will do it."* |
| [P89, M] | 5.25 | *"I think it is interesting that I can schedule exercise plans and how much I need to exercise when my schedule is busy and how I can fit it in."* | COOP = 7 | *"I would feel that someone is on my side, we would push each other"* |
| [P18, F] | 6.5 | *"Calories burned"* | CMPT = 6.5 | *"If feasible, I would certainly try to get into the top 3"* |
| [P22, F] | 5.25 | *"step counter"* | CMPT = 5.5 | *"gives me a benchmark to work against"* |
| [P161, M] | 6 | *"to provide food opinion"* | SCOMP = 5.5 | *"comparing with others motivate to do more."* |
| [P67, M] | 7 | *"Reminder for regular exercise"* | SCOMP = 6 | *"It will create competition that will influence peoples decision to do more"* |
| [P2, F] | 5 | *"Reminder to engage in physical activities, features like something explaining how fit I am and how I can get better, features that can link me with others like me or people that can encourage me to be fit."* | SLEARN = 6 | *"I can see what others are doing and that can motivate me"* |
| [P12, M] | 5.25 | *"A key feature would be to be able to track my workout and input sets x repetitions for even somewhat unconventional workouts"* | SLEARN = 5.25 | *"I am fairly competitive and being able to see what my friends are doing may help push me even further to accomplish my goals"* |

**Table 9.** Sample comments on the relationship between perceived usefulness and non-receptiveness to persuasive features. GST/SMT = Goal-Setting/Self-Monitoring, REWD = Rewards, COOP = Cooperation, CMPT = Competition, SCOMP = Social Comparison, SLEARN = Social Learning.

| No. | Usefulness | Key Feature | | Persuasive Feature |
|---|---|---|---|---|
| [P96, M] | 1.5 | *"a way to connect with friends and see their progress"* | GST/SMT = 1 | *"I don't track calories and that feature would be useless to me"* |
| [P43, F] | 1.5 | *"Diet diary allowing for tracking of ketogenic diets."* | GST/SMT = 2 | *"I just don't think I could live up to the goals and that would depress me"* |
| [P80, F] | 1.25 | *"Exercise tips"* | REWD = 1 | *"I don't need rewards."* |
| [P6, M] | 2.25 | *"Stopwatch"* | REWD = 1 | *"I personally don't care about meaningless point systems"* |
| [P59, F] | 2 | *"online exercise videos"* | COOP = 1 | *"I don't like team work."* |
| [P50, F] | 2.5 | *"step counter"* | COOP = 1 | *"I do not like to connect with friends on applications"* |
| [P75, M] | 2.75 | *"A way to count calories by entering what I ate for the day"* | CMPT = 2 | *"I don't believe in fitness competitions."* |
| [P59, F] | 2 | *"online exercise videos"* | CMPT = 1 | *"I don't like competition/comparison."* |
| [P60, M] | 2.75 | *"A way to count calories by entering what I ate for the day"* | SCOMP = 1.5 | *"I don't believe in fitness challenges. Much of fitness is genetic and therefore, unfair."* |
| [P103, F] | 1 | *"Exercise selection according to skill and abilities or disabilities"* | SCOMP = 1 | *"Don't have any intention to compare myself to others."* |
| [P98, M] | 1.75 | *"Diet Tracker"* | SLEARN = 1 | *"I wouldn't share my data or want to know how well or bad someone else is doing"* |
| [P43, F] | 1.5 | *"Diet diary allowing for tracking of ketogenic diets"* | SLEARN = 1 | *"I don't care what others achieve"* |

**Table 10.** Sample comments on the relationship between perceived aesthetics and receptiveness to Competition and Social Learning, CMPT = Competition, SLEARN = Social Learning.

| No. | Aesthetics | Key Feature | | Persuasive Feature |
|---|---|---|---|---|
| [P22, F] | 6.17 | *"step counter"* | CMPT = 5.5 | *"gives me a benchmark to work against"* |
| [P14, F] | 5.12 | *"I would like the app to remind me to drink water."* | CMPT = 7 | *"this is another good way to motivate me to use the app"* |
| [P23, F] | 5.5 | *"Videos that you can follow along and exercise to with increasing difficulty such as strength yoga or treadmill workouts"* | SLEARN = 6 | *"I am pretty competitive I found that when I had friends challenge me in the past it motivated me"* |
| [P26, M] | 7 | *"Calorie tracker"* | SLEARN = 7 | *"Social pressure helps one to work out"* |

**Table 11.** Sample comments on the relationship between perceived aesthetics and non-receptiveness to Competition and Social Learning, CMPT = Competition, SLEARN = Social Learning.

| No. | Aesthetics | Key Feature | | Persuasive Feature |
|---|---|---|---|---|
| [P103, F] | 1 | *"Exercise selection according to skill and abilities or disabilities"* | CMPT = 1 | *"I'm not the competitive type. I don't do things to be better than others."* |
| [117, M] | 3 | *"Stretching exercises"* | CMPT = 2 | *"Do not like the idea of competing"* |
| [P50, F] | 2.5 | *"step counter"* | SLEARN = 1 | *"I do not enjoy comparing my excercise to others because I only feel worse"* |
| [P7, M] | 2 | *"Easy to use"* | SLEARN = 1 | *"For me, fitness is a personal goal, I don't need to compare myself"* |

## 5. Discussion

We have presented path models showing the relationship between the perceived UX design attributes of a fitness app and users' receptiveness to commonly employed persuasive features. The perceived UX design attributes include *perceived aesthetics*, *perceived usability*, *perceived credibility*, and *perceived usefulness*. On the other hand, the persuasive features include Goal-Setting/Self-Monitoring, Reward, Competition, Cooperation, Social Comparison, and Social Learning as well as the overall *perceived persuasiveness* of the fitness app. The goodness of fit (GOF) of the models ranges from 0.70 (for Cooperation and Social Comparison) to 0.78 (for the overall *perceived persuasiveness* of the fitness app). These GOF values are high, indicating that the respective models fit their empirical data to a large degree. According to Hussain et al. [43], a GOF value of 0.10, 0.25, and 0.36 is an indication that the overall validation of a given model by its empirical data is small, medium, and large, respectively. Moreover, with regard to $R^2$ value, the model shown in Figure 3 accounts for 72% of the variance of the overall *perceived persuasiveness* of the fitness app, with *perceived usefulness* ($\beta = 0.68$, $p < 0.001$) and *perceived aesthetics* ($\beta = 0.25$, $p < 0.01$) explaining most of the variance. According to Sanchez [14], $R^2$ values above 60% are considered high values, those between 60% and 30% are considered moderate, and those less than 30% are considered low. Thus, the coefficient of determination of the overall *perceived persuasiveness* of the fitness app prototype is high. Moreover, three of the models shown in Figure 4 account for more than 30% of the variance of users' receptiveness to three persuasive features (indicating moderate $R^2$ values), while the other three account for less than 30% (indicating low $R^2$ values). Specifically, regarding Goal-Setting/Self-Monitoring, 58% of its variance is explained by mostly *perceived usefulness* ($\beta = 0.56$, $p < 0.001$). Secondly, regarding Reward, 39% of its variance is explained by mostly *perceived usefulness* ($\beta = 0.53$, $p < 0.001$). Thirdly, regarding Social Learning, 33% of its variance is explained by mostly *perceived usefulness* ($\beta = 0.46$, $p < 0.001$) and *perceived aesthetics* ($\beta = 0.22$, $p < 0.05$).

Overall, the PLSPM shows that the relationship between *perceived usefulness* and overall *perceived persuasiveness* ($\beta = 0.68$, $p < 0.001$) is the strongest, followed by the relationship

between *perceived aesthetics* and overall *perceived persuasiveness* (β = 0.25, *p* < 0.01). This suggests that *perceived usefulness* (the belief that a fitness app will help users to accomplish their health goal) is the most important design attribute of a persuasive application aimed at behavior change in the health domain. The second most important design attribute is *perceived aesthetics* (the belief that the perceived UX design of a persuasive system is visually appealing and pleasing) in the design of persuasive applications aimed at behavior change. One explanation for why both *perceived usefulness* and *perceived aesthetics* have a significant positive relationship with two or more of the six persuasive features (as shown in Table 7) is that there is an underlying positive relationship between both perceived UX design attributes. For example, in prior studies in the health domain, Oyibo et al. [9] and Van der Heijden [10] found that the relationship between both perceived UX design attributes is significantly strong.

### 5.1. Relationship between Perceived Usefulness and Receptiveness to Persuasive Features

In our PLSPM, we found that *perceived usefulness* has the strongest and most consistent relationship with users' receptiveness to persuasive features, as shown in Figures 3 and 4 and Table 7. This suggests that the higher users perceive a fitness application as useful, the more likely they are to perceive it as persuasive and be receptive to its persuasive features, such as Goal-Setting/Self-Monitoring, Reward, and Cooperation. For example, P49, whose average rating of the *perceived usefulness* of the fitness app is high (M = 5.5), rated her level of receptiveness to the Cooperation feature as high as well (M = 7, which is above the neutral value of 4). This suggests her high perception of the *usefulness* of the fitness app correlated with her high level of receptiveness to the Cooperation feature (strategy) of encouraging behavior change. P49's receptiveness to the Cooperation feature is also evident in her comment, "*If someone is relying on me then I will do it.*" Secondly, P161, whose average rating of *perceived usefulness* is high (M = 6), rated his receptiveness to the Comparison feature as high as well (M = 5.5). Specifically, he commented, "*comparing with others motivate to do more.*" Thirdly, P216, whose average rating of the *perceived usefulness* of the fitness app is high (M = 6.5), rated her receptiveness to the Reward feature as high as well (M = 7). This suggests that her high perception of the fitness app as useful correlated with her high level of receptiveness to the *Reward* feature of motivating behavior change. Specifically, regarding the Reward storyboard, P216 commented thus, "... *I would want to reach the goal and earn the points.*"

Given the all-positive relationships between *perceived usefulness* and receptiveness to persuasive features, as shown in Tables 8 and 9, one may want to know what it means for a fitness app to be "useful." Simply put, the *perceived usefulness* of a fitness app can be regarded as the belief that the app will help users accomplish and/or reach their goals easily and improve their exercise performance ultimately (see the questionnaire items on *perceived usefulness* in Table 3, for example). In other words, a useful fitness app should be able to support persuasive features that will help users accomplish their exercise goals easily. Such persuasive features include Goal-Setting, Self-Monitoring (e.g., calorie counter, step counter, and activity tracking), Reminder, and Social Support. The following are some of the participants' comments on one key feature they expected the fitness app (presented in Figure 1) to have if they were to use it:

1. "*The ability to define goals*" [P61].
2. "*I would expect the app to tell me how well I have been doing with my fitness goals*" [P183].
3. "*A calorie counter to help you lose weight and keep track of the foods you eat*" [P203].
4. "*I would expect it to have a way to track a lot of information without having to manually enter them*" [P104].
5. "*The most important key feature would have to be being able to schedule my routine and having reminders as well*" [P96].
6. "*Consistent reminder from the app to work out and motivational quotes or pics to follow with reminder*" [P84].
7. "*Reminders about when sessions should be done*" [P42].

8. *"I would expect it to allow me to change my goals as my needs change [during] the day"* [P29].
9. *"The ability to add friends for support"* [P152].
10. *"A way to connect with friends and see their progress"* [P33].

### 5.2. Relationship between Perceived Aesthetics and Receptiveness to Persuasive Features

Apart from *perceived usefulness*, we found that *perceived aesthetics* has a positive relationship with *perceived persuasiveness* ($\beta$ = 0.25, $p < 0.01$) and users' receptiveness to two persuasive features: Competition ($\beta$ = 0.30, $p < 0.05$) and Social Learning ($\beta$ = 0.22, $p < 0.05$). This suggests that overall, the higher users' perception of a fitness application as aesthetic, the higher they tend to perceive it as persuasive. Particularly, the higher they will be receptive to its Competition and Social Learning features. For example, P23, whose average rating of *perceived aesthetics* was high (M = 5.67), rated their receptiveness to the Competition feature as high as well (M = 6.75). This suggests their perception of the fitness app as highly aesthetic correlated with their receptiveness to the Competition feature (strategy) of encouraging behavior change. Thus, P23 commented, "*Competition is a good way of inducing exercise, and I think I would enjoy an app that had this aspect.*" Similarly, P69, whose average rating of *perceived aesthetics* was high (M = 5.33), rated their receptiveness to the Social Learning feature as high as well (M = 5). Additionally, they commented thus, " . . . *this would work to motivate and influence me to not slack off or take unnecessary time away from physical activity.*" In a nutshell, the positive relationship between *perceived aesthetics*, on one hand, and *perceived persuasiveness and* receptiveness to Competition and Social Learning, on the other hand, confirmed the extant theory and finding that a visually attractive persuasive system is more likely to persuade users than a less attractive one [1,6,35].

### 5.3. Gender Differences

Overall, the gender-based submodels are similar to the models for the entire dataset (see Table 6). In particular, regardless of gender, the relationship between *perceived usefulness* and each strategy is significant ($p < 0.05$) except for Competition for males. That said, the only significant difference between males and females is in the relationship between *perceived credibility* and Cooperation ($p < 0.01$). However, neither the path coefficient for males ($\beta$ = 0.11, $p$ = n.s) nor that for females ($\beta$ = $-0.06$, $p$ = n.s) is significant. Although there is a significant difference between both genders, neither of the path coefficients in the male and female submodels (just as in the overall model) are significant. Moreover, there is a marginal significant difference between both genders with respect to the relationship between *perceived usefulness* and Competition ($p$ = 0.051), with the path coefficient for females ($\beta$ = 0.45, $p < 0.001$) being stronger than that for males ($\beta$ = 0.18, $p$ = n.s). This finding indicates that the higher females perceive a fitness app as useful, the more likely they are to be receptive to the Cooperation feature. However, this is not the case for males. Similarly, there is a marginal significant difference between both genders with respect to the relationship between *perceived credibility* and Social Learning ($p$ = 0.055), with the path coefficient for males ($\beta$ = 0.20, $p < 0.05$) being stronger than that for females ($\beta$ = 0.02, $p$ = n.s). This finding indicates that the higher males perceive a fitness app as credible, the more likely they are to be receptive to the Social Learning feature. However, this is not the case for females.

### 5.4. Summary of Findings

In this section, we summarize our key findings, beginning with the relationships that have to do with *perceived usability* and *perceived credibility*. While the relationship between *perceived usefulness/perceived aesthetics* and receptiveness to persuasive features are positive, our PLSPM shows that *perceived usability* has a negative relationship with receptiveness to two of the persuasive features: Cooperation ($\beta$ = $-0.15$, $p < 0.05$) and Social Learning ($\beta$ = $-0.15$, $p < 0.05$). However, the relationships are not strong to the extent to conclude that *perceived usability* has a negative impact on users' receptiveness to persuasive features, which contrasts the seventh postulate in the PSD model, which holds that an easy-to-use

persuasive system is more likely to be persuasive [6]. Based on Chin's guideline [42], path coefficients should be at least equal to 0.2 to be considered relevant. Thus, given that five of the seven relationships between *perceived usability* and receptiveness to persuasive features are statistically non-significant and two are relatively weak (with absolute value less than 0.2), we conclude that *perceived usability*, similar to *perceived credibility*, has little or no significant positive relationship with users' receptiveness to persuasive features. Thus, overall, based on the significant relationships between the perceived UX design constructs, on one hand, and the overall *perceived persuasiveness* and persuasive features, on the other hand, we summarize the main results of the study as follows:

1.  The higher the *perceived usefulness* of a fitness app by potential users is, the more likely they are to be persuaded by the app and its key persuasive features such as Goal-Setting/Self-Monitoring, Reward, Competition, Cooperation, Social Comparison, and Social Learning.
2.  The higher the *perceived aesthetics* of a fitness app by potential users is, the more likely they are to be persuaded by the app and its key persuasive features such as Competition and Social Learning.

The above findings are in line with our prior finding in the context of a Persuasive Technology Acceptance Model for a fitness application [9]. In that study, we found that *perceived usefulness* and *perceived aesthetics*, regardless of culture (individualist and collectivist), are the strongest perceived UX-design predictors of users' *intention to use* a fitness app to motivate behavior change. Hence, based on the current and prior findings, we recommend that persuasive applications such as fitness apps should be designed to be visually *attractive* and functionally *useful* to meet users' hedonic needs and utilitarian needs, respectively. Specifically, *perceived aesthetics* can be said to cater to users' emotional (hedonic) needs, while *perceived usefulness* can be said to cater to their instrumental (utilitarian) needs such as setting goals and monitoring their progress over time and being able to compete or cooperate with others.

*5.5. Limitations*

The study has a number of limitations. The first and foremost limitation of the study is that our findings are based on perception, a prototyped fitness app, and storyboards of low fidelity. Thus, our findings may not generalize to an actual setting in which participants in the form of users have to interact with an actual fitness app. For this reason, we recommend that future studies use higher-fidelity prototypes to improve the ecological validity of the findings. The second limitation of the study is that we did not investigate the moderating effect of key demographic variables such as age, and education on the findings. In future work, we aim to address this limitation. The third limitation of the study is that its findings are based on a single prototype of a fitness app rather than multiple or different prototypes that cut across different domains. This limits the generalizability of the current findings. Hence, further studies, especially based on higher-fidelity and multiple-domain app prototypes and persuasive-feature storyboards, ought to be carried out in the future to investigate whether the current findings generalize to other domains such as healthy eating, smoking cessation, and energy conservation. The fourth limitation of the study is that most of the participants are Canadians and Americans resident in North America. This may also affect the generalizability of our findings to other demographics in other continents. Therefore, in our future work, we intend to verify our findings in the context of an actual system. Secondly, we intend to extend our study to other demographics outside North America to verify the generalizability of our findings.

*5.6. Contributions*

The main contribution of our paper is that it serves as exploratory work and presents some preliminary findings on the hypothesized relationship between UX design attributes (such as perceived usefulness and perceived aesthetics) and users' receptiveness to the persuasive features of a persuasive application (e.g., Goal-Setting/Self-Monitoring, Reward,

and Competition). Future work can build on these preliminary findings. Particularly, due to the methodological limitation of the paper (e.g., the use of a single prototype rather than multiple cutting across different domains), we recommend further studies be carried out to test the generalizability of the current findings in the fitness domain to other domains such as healthy eating, smoking cessation, and energy conservation.

## 6. Conclusions

We have presented a path model to understand the relationship between perceived UX design attributes and users' receptiveness to persuasive features implemented in persuasive technologies. Specifically, we examined the relationship between *perceived aesthetics*, *perceived usability*, *perceived credibility*, and *perceived usefulness* of a fitness app (on one hand) and users' receptiveness to its overall *perceived persuasiveness* and commonly employed persuasive features (on the other hand). We focus on persuasive features such as Goal-Setting/Self-Monitoring, Reward, Cooperation, Competition, Social Comparison, and Social Learning, which are drawn from the PSD model [6] and illustrated by storyboards. The results of our PLSPM show that *perceived usefulness* has the strongest relationship with users' receptiveness to the overall *perceived persuasiveness* of the fitness app and all of its persuasive features. Moreover, *perceived aesthetics* has the second strongest relationship with users' receptiveness to the overall *perceived persuasiveness* of the fitness app and two of its socially oriented persuasive features (Competition and Social Learning). However, *perceived usability* and *perceived credibility* have a relatively weak or no significant relationship with users' receptiveness to the overall *perceived persuasiveness* of the fitness app and its persuasive features. Based on our findings, we recommended that designers of PTs, especially for behavior change in the health domain, should focus on a visually attractive UX design, which possesses persuasive features, such as Goal-Setting/Self-Monitoring, Reminder, Reward, and Social Support, which users care about and will help them accomplish their health goals easily.

**Author Contributions:** Conceptualization, K.O. and J.V.; methodology, K.O.; software, NA; validation, K.O. and J.V.; formal analysis, K.O.; investigation, K.O.; resources, J.V.; data curation, K.O.; writing—original draft preparation, K.O.; writing—review and editing, K.O. and J.V.; visualization, K.O.; supervision, J.V.; project administration, J.V.; funding acquisition, J.V. All authors have read and agreed to the published version of the manuscript.

**Funding:** This research was funded by Julita Vassileva's Natural Sciences and Engineering Research Council of Canada Discovery Grant (RGPIN-2016-05762).

**Institutional Review Board Statement:** The study was conducted according to the guidelines of the Declaration of Helsinki, and approved by the Institutional Review Board (or Ethics Committee) of University of Saskatchewan (protocol code BEH 17-362) on 19 October 2017.

**Informed Consent Statement:** Informed consent was obtained from all subjects involved in the study.

**Data Availability Statement:** The data is available upon request from the authors.

**Conflicts of Interest:** The authors declare no conflict of interest. The funders had no role in the design of the study; in the collection, analyses, or interpretation of data; in the writing of the manuscript, or in the decision to publish the results.

## Appendix A

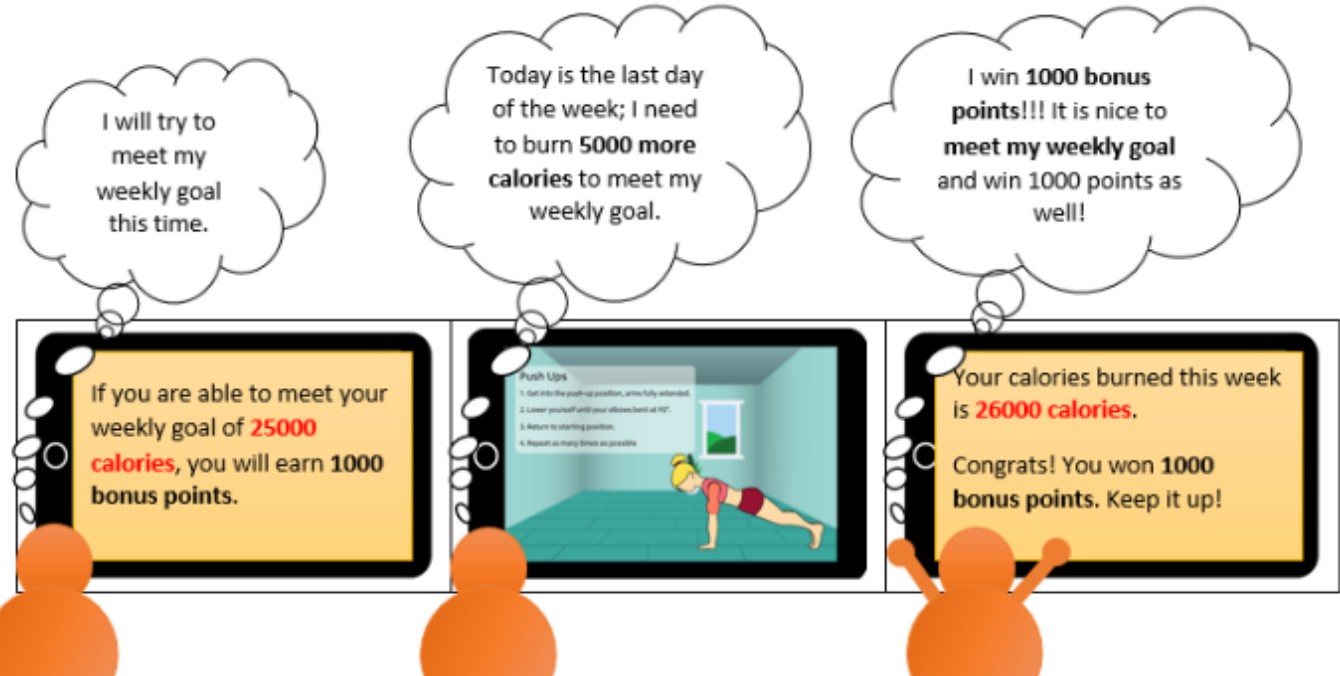

**Figure A1.** Low-fidelity storyboard illustrating the Goal-Setting/Self-Monitoring feature of the fitness app [30].

**Figure A2.** Low-fidelity storyboard illustrating the Reward feature of the fitness app [30].

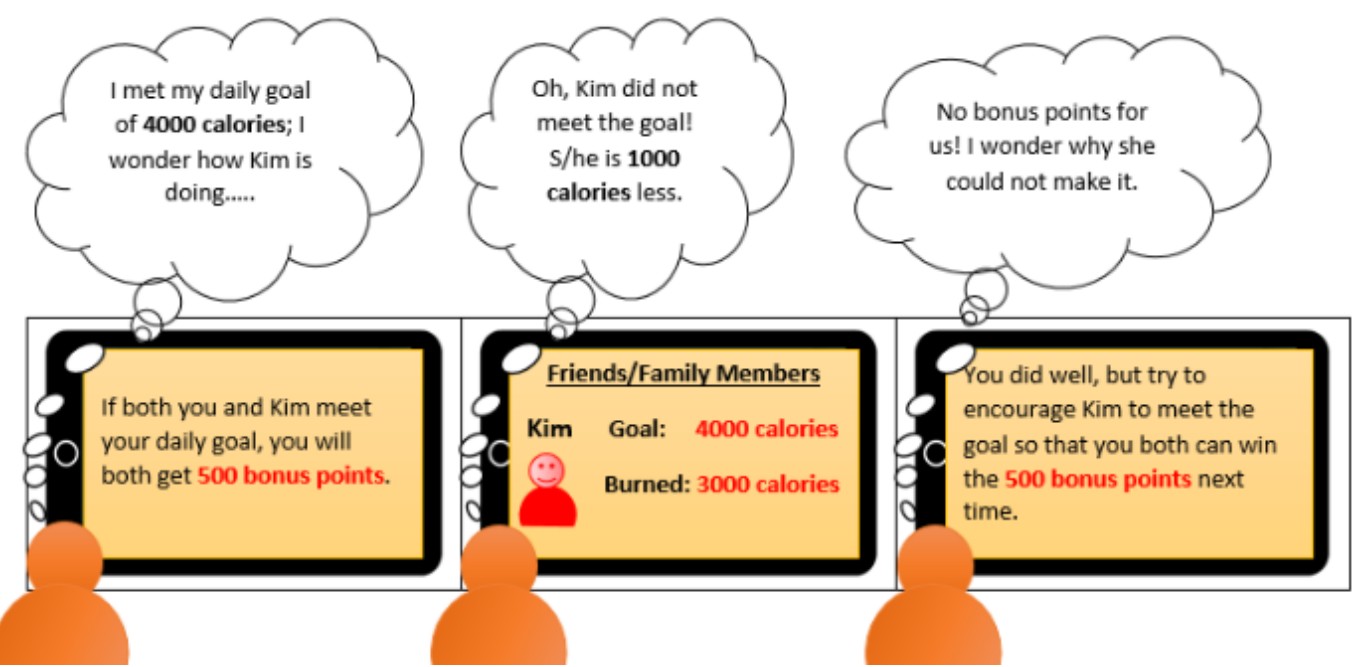

**Figure A3.** Low-fidelity storyboard illustrating the Cooperation feature of the fitness app [30].

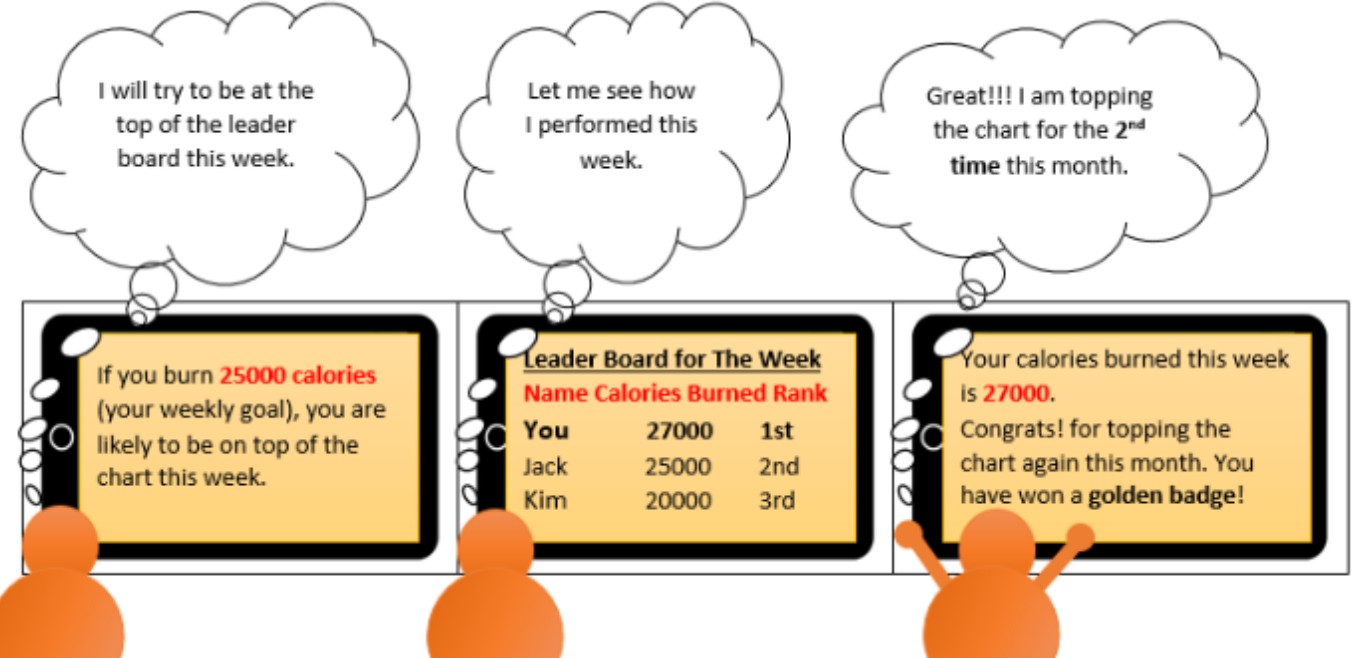

**Figure A4.** Low-fidelity storyboard illustrating the Competition feature of the fitness app [30].

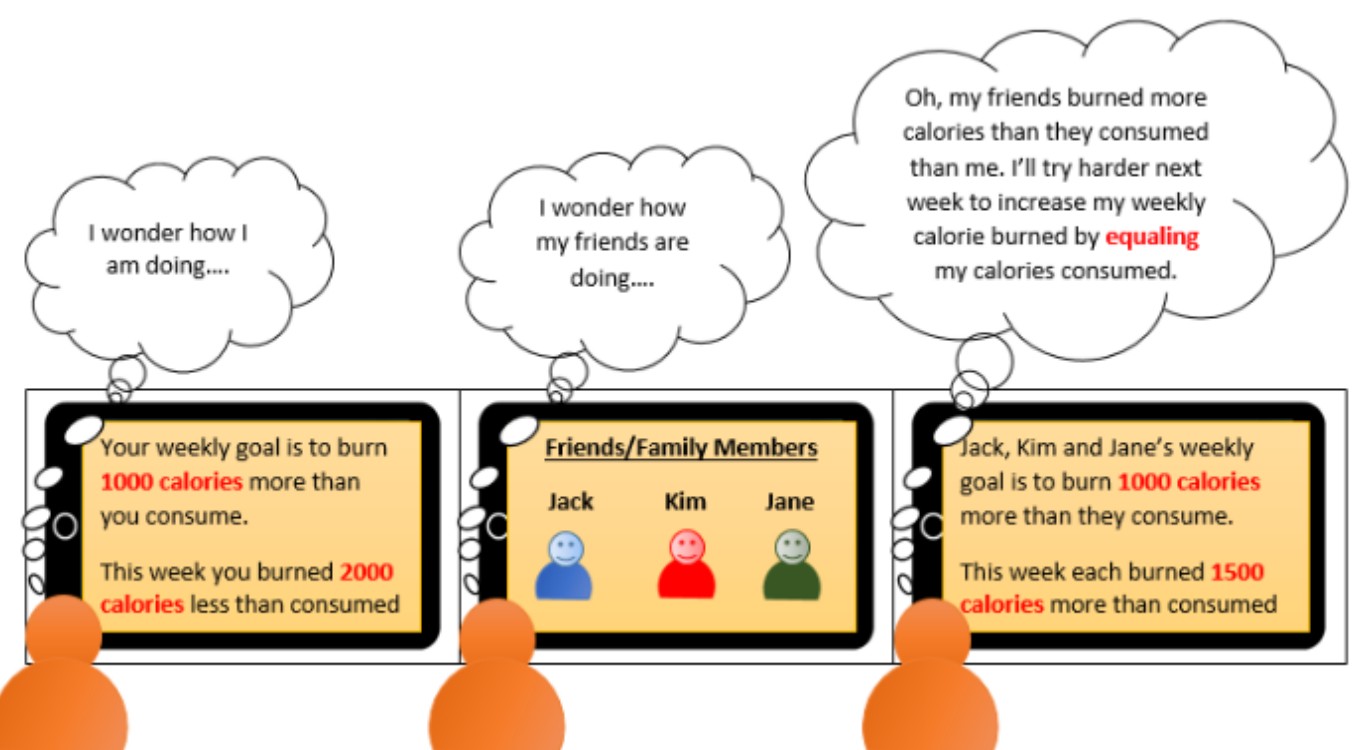

**Figure A5.** Low-fidelity storyboard illustrating the Social Comparison feature of the fitness app [30].

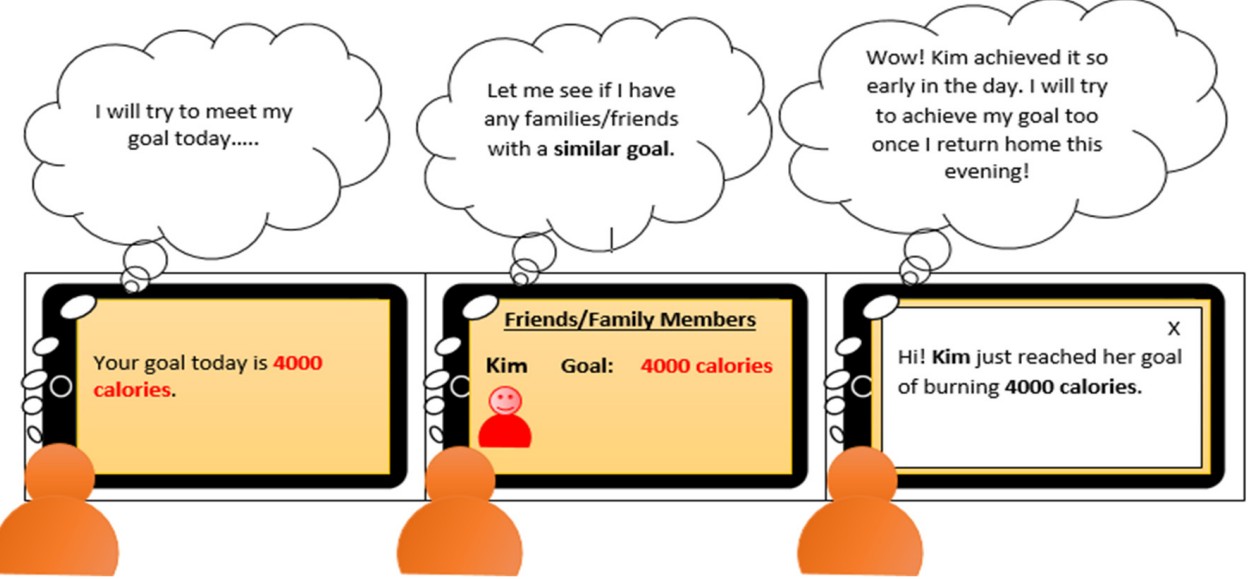

**Figure A6.** Low-fidelity storyboard illustrating the Social Learning feature of the fitness app [30].

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
