# Peer review of "Relationship between Perceived UX Design Attributes and Persuasive Features: A Case Study of Fitness App"

_information, doi:10.3390/info12090365_

Round 1

Reviewer 1 Report

The paper addresses a very interesting and relevant topic, in particular how the UX design attributes perceived aesthetics, perceived usability, perceived credibility and perceived usefulness affect users' receptiveness to persuasive features. The paper is very well written, and provides a clear overview of relevant literature.

The key issue I have with the paper is: Whereas the paper does describe a solid analysis of data, resulting in interesting findings, I do question the generalizability of the results. In particular, the authors have created one single stimulus (one design of a fictive app, in combination with a series of storyboards). Participants are asked to judge this single stimulus. In other words - the authors have studied in detail how this particular stimulus is experienced, and how the user experiences of individual elements affect the overarching construct of receptiveness to persuasive feature. 

The paper does however suggest in the introduction, the discussion and the conclusion that the findings can be generalized (the perceived usefulness of a fitness application (...) can be associated with (...)". In order to make this statement, I would like to see how different stimuli (e.g. different prototypes with different levels of perceived aesthetics, usability, credibility and usefulness) result in different levels of users' receptiveness to persuasive features, and how these findings an be combined in a single model.

The authors might frame the current study as a case study, and avoid general statements on the findings. In this case study, the authors might indicate that for this particular design, certain findings can be concluded.

I would however suggest to re-run the study with multiple stimuli, resulting in generalizible findings.

Author Response

File uploaded.

Reviewer 2 Report

Authors present a study aimed to investigate the relationship between perceived UX design attributes and users' receptiveness to commonly employed persuasive features of a fitness application. They conducted an exploratory study through an online survey. 

The models used to analyse the relationships among the UX design attributes and other variables/features are quite simple and appropriate. 

The manuscript is very well structured and quite clear and understandable. 

Authors conducted a high quality research and results are quite interesting. I have only a few minor comments:

1. Perhaps, the scales (quatitative questions) used in the survey could be better explained. It is not really clear what values a participant could assign to each item.

2. There is a typo in the lines number 381 and number 385-385: "Table 10 and Table 10" and "On the other hand, Table 10 is for..." the right number should be "11"

Finally, I would like to ask about a personal curiosity: Why did authors not conduct an analysis of relationships among age and the other variables? Perhaps, this analysis was performed in the previous authors' studies, but I am not sure. 

Congratulations

Author Response

File uploaded.

Reviewer 3 Report

This paper proposes an interesting research agenda and is well written.

The authors tackle an interesting challenge in UX/IxD Design, namely the understanding of the relationship between user UX design attributes and users’ receptiveness to the persuasive features of a persuasive technology aimed at motivating behavior change. Research took place in US & Canada and involved 228 participants interacting with a prototype of a fitness application.

Although the paper is carefully written, the research methodology is well established, and the use of UX evaluation techniques and statistical analysis are all carefully executed and clear, my major concern is about the prototype and the validity of its actual use and evaluation. The authors themselves identify this problem of this study (5.6). In my opinion the use of storyboards of a very low-fidelity prototype pose a number of weaknesses regarding the validity of evaluation and the generalisation of actual findings. For such a study, I would expect at least a medium (mockup) to high fidelity prototype.

On the other hand the approach is interesting and provides to the UX/IxD community a method for relating UX design attributes to users' perceived persuasiveness.

My proposal is:

- to explicitly state (abstract & main text) that the research was based on a low-fidelity prototype in the form of storyboard.

- to give more emphasis on the presentation of the methodology and less on the actual results. After all their validity is limited by the limitations posed by the prototype.

Make the following minor changes:

- table 1 fix: "the beilef that"

- table 1 fix : "It is the belief that a persuasive system will be easy to use, understand and free of efforts."

- table 2 fix:  "the capacity of a peruasive system"

- line 161-162: add reference

- line 251: you state that "Wrong responses were treated as missing data points and replaced by their respective average scores during the data analysis" Please explain how this has been done, it is not clear.

- Table 7: you use abbreviations for some terms and not for others. Please be consistent.

Author Response

File uploaded.

Round 2

Reviewer 1 Report

The authors have considered the feedback from the reviewers, and have made minor changes to the manuscript. In my view, however, the key underlying methodological challenge has not been properly addressed.

Whereas the paper does describe a solid analysis of data, resulting in interesting findings, I still question the generalizability of results. Opposed to the feedback from the authors on the first review, the generalizability is not particularly related to the domain (fitness vs. other domains), but about stimuli. Participants have now evaluated 1 single prototype. In other words, only 1 datapoint has been evaluated. In order to be able to make statements on relation between different levels of perceived aesthetics, usability, credibility and usefulness and receptiveness to persuasive features, multiple prototypes should be compared. For example, three versions of the fitness app, with different levels of aesthetics, usability etc. Without comparing these different stimuli, the author will not be able to make a claim regarding relation between underlying factors and receptiveness.

I strongly think that this methodological issue currently blocks the acceptability of the manuscript. And this cannot be fixed by adding a sentence to the limitations section. 

Moreover, the authors still claim "Our main contribution to knowledge is that we showed, in a replicated fashion, that the perceived usefulness of a fitness application such as a fitness application can be associated with users’ receptiveness to the persuasive features with which it is equipped.". I assume they mean "(...) the perceived usefulness of a persuasive application such as a fitness application (...)". Given the problem of the "1 data point" methodology, one cannot say "the usefulness of a fitness application (...) can be associated with users' receptiveness".

In summary:

  • even though the paper is now framed as a case study, the authors still make general statements regarding the findings.
  • the underlying methodology is not sound. The high number of participants suggests a solid study, but the stimuli used in the study (a single prototype) is not sufficient for the 'generic findings' the authors present.
  • whereas the authors can reframe the study (presenting the findings without generic claims), I think the methodology used will suggest a level of validity which is not true, given the limitations in the stimuli.

Author Response

File uploaded.
